# Discovering Potential Correlations via Hypercontractivity

**Hyeji Kim**[1*]    **Weihao Gao**[1*]    **Sreeram Kannan**[2†]    **Sewoong Oh**[1‡]    **Pramod Viswanath**[1*]

University of Illinois at Urbana Champaign[1] and University of Washington[2]

{hyejikim,wgao9}@illinois.edu,ksreeram@uw.edu,{swoh,pramodv}@illinois.edu

## Abstract

Discovering a correlation from one variable to another variable is of fundamental scientific and practical interest. While existing correlation measures are suitable for discovering *average* correlation, they fail to discover hidden or *potential* correlations. To bridge this gap, (i) we postulate a set of natural axioms that we expect a measure of potential correlation to satisfy; (ii) we show that the *rate* of information bottleneck, i.e., the *hypercontractivity* coefficient, satisfies all the proposed axioms; (iii) we provide a novel estimator to estimate the hypercontractivity coefficient from samples; and (iv) we provide numerical experiments demonstrating that this proposed estimator discovers potential correlations among various indicators of WHO datasets, is robust in discovering gene interactions from gene expression time series data, and is statistically more powerful than the estimators for other correlation measures in binary hypothesis testing of canonical examples of potential correlations.

## 1  Introduction

Measuring the strength of an association between two random variables is a fundamental topic of broad scientific interest. Pearson's correlation coefficient [1] dates from over a century ago and has been generalized seven decades ago as maximal correlation (mCor) to handle nonlinear dependencies [2–4]. Novel correlation measures to identify different kinds of associations continue to be proposed in the literature; these include maximal information coefficient (MIC) [5] and distance correlation (dCor) [6]. Despite the differences, a common theme of measurement of the empirical *average* dependence unites the different dependence measures. Alternatively, these are *factual* measures of dependence and their relevance is restricted when we seek a *potential* dependence of one random variable on another. For instance, consider a hypothetical city with very few smokers. A standard measure of correlation on the historical data in this town on smoking and lung cancer will fail to discover the fact that smoking causes cancer, since the average correlation is very small. On the other hand, clearly, there is a potential correlation between smoking and lung cancer; indeed applications of this nature abound in several scenarios in modern data science, including a recent one on genetic pathway discovery [7].

Discovery of a potential correlation naturally leads one to ask for a measure of potential correlation that is statistically well-founded and addresses practical needs. Such is the focus of this work, where our proposed measure of potential correlation is based on a novel interpretation of the *Information Bottleneck* (IB) principle [8]. The IB principle has been used to address one of the fundamental tasks in supervised learning: given samples $\{X_i, Y_i\}_{i=1}^n$, how do we find a *compact* summary of a variable

---

[*]Coordinated Science Lab and and Department of Electrical and Computer Engineering
[†]Department of Electrical Engineering
[‡]Coordinated Science Lab and Department of Industrial and Enterprise Systems Engineering

$X$ that is most *informative* in explaining another variable $Y$. The output of the IB principle is a compact summary of $X$ that is most relevant to $Y$ and has a wide range of applications [9, 10].

We use this IB principle to create a measure of correlation based on the following intuition: if $X$ is (potentially) correlated with $Y$, then a relatively compact summary of $X$ can still be very informative about $Y$. In other words, the maximal ratio of how informative a summary can be in explaining $Y$ to how compact a summary is with respect to $X$ is, conceptually speaking, an indicator of potential correlation from $X$ to $Y$. Quantifying the compactness by $I(U; X)$ and the information by $I(U; Y)$ we consider the *rate of information bottleneck* as a measure of potential correlation:

$$s(X;Y) \equiv \sup_{U-X-Y} \frac{I(U;Y)}{I(U;X)} \,, \tag{1}$$

where $U - X - Y$ forms a Markov chain and the supremum is over all summaries $U$ of $X$. This intuition is made precise in Section 2, where we formally define a natural notion of potential correlation (Axiom 6), and show that the rate of information bottleneck $s(X; Y)$ captures this potential correlation (Theorem 1) while other standard measures of correlation fail (Theorem 2).

This ratio has only recently been identified as the *hypercontractivity* coefficient [11]. Hypercontractivity has a distinguished and central role in a large number of technical arenas including quantum physics [12, 13], theoretical computer science [14, 15], mathematics [16–18] and probability theory [19, 20]. In this paper, we provide a novel interpretation to the hypercontractivity coefficient as a measure of potential correlation by demonstrating that it satisfies a natural set of axioms such a measure is expected to obey.

For practical use in discovering correlations, the standard correlation coefficients are equipped with corresponding natural sample-based estimators. However, for hypercontractivity coefficient, estimating it from samples is widely acknowledged to be challenging, especially for continuous random variables [21–23]. There is no existing algorithm to estimate the hypercontractivity coefficient in general [21], and there is no existing algorithm for solving IB from samples either [22, 23]. We provide a novel estimator of the hypercontractivity coefficient – the first of its kind – by bringing together the recent theoretical discoveries in [11, 24] of an alternate definition of hypercontractivity coefficient as ratio of Kullback-Leibler divergences defined in (5), and recent advances in joint optimization (the max step in Equation 1) and estimating information measures from samples using importance sampling [25].

Our **main contributions** are the following:

- We postulate a set of natural axioms that a measure of potential correlation from $X$ to $Y$ should satisfy (Section 2).

- We show that $\sqrt{s(X;Y)}$, our proposed measure of potential correlation, satisfies all the axioms we postulate. In comparison, we prove that existing standard measures of correlation not only fail to satisfy the proposed axioms, but also fail to capture canonical potential correlations captured by $\sqrt{s(X;Y)}$ (Section 2). Another natural candidate is mutual information, but it is not clear how to interpret the value of mutual information as it is unnormalized, unlike all other measures of correlation which are between zero and one.

- Computation of the hypercontractivity coefficient from samples is known to be a challenging open problem. We introduce a novel estimator to compute hypercontractivity coefficient from i.i.d. samples in a statistically consistent manner for continuous random variables, using ideas from importance sampling and kernel density estimation (Section 3).

- In a series of synthetic experiments, we show empirically that our estimator for the hypercontractivity coefficient is statistically more powerful in discovering a potential correlation than existing correlation estimators; a larger power means a larger successful detection rate for a fixed false alarm rate (Section 4.1).

- We show applications of our estimator of hypercontractivity coefficient in two important datasets: In Section 4.2, we demonstrate that it discovers hidden potential correlations among various national indicators in WHO datasets, including how aid is potentially correlated with the income growth. In Section 4.3, we consider the following gene pathway recovery problem: we are given samples of four gene expressions time series. Assuming we know that gene A causes B, that B causes C, and that C causes D, the problem is to discover that

these causations occur in the sequential order: A to B, and then B to C, and then C to D. We show empirically that the estimator of the hypercontractivity coefficient recovers this order accurately from a vastly smaller number of samples compared to other state-of-the art causal influence estimators.

## 2 Axiomatic approach to measure potential correlations

We propose a set of axioms that a measure of potential correlation should satisfy and propose a new measure of correlation that satisfies all the proposed axioms.

**Axioms for potential correlation.** We postulate that a *measure of potential correlation* $\rho^* : \mathcal{X} \times \mathcal{Y} \to [0, 1]$ between two random variables $X \in \mathcal{X}$ and $Y \in \mathcal{Y}$ should satisfy:

1. $\rho^*(X, Y)$ is defined for any pair of non-constant random variables $X$ and $Y$.

2. $0 \leq \rho^*(X, Y) \leq 1$.

3. $\rho^*(X, Y) = 0$ iff $X$ and $Y$ are statistically independent.

4. For bijective Borel-measurable functions $f, g : \mathbb{R} \to \mathbb{R}$, $\rho^*(X, Y) = \rho^*(f(X), g(Y))$.

5. If $(X, Y) \sim \mathcal{N}(\mu, \Sigma)$, then $\rho^*(X, Y) = |\rho|$, where $\rho$ is the Pearson correlation coefficient.

6. $\rho^*(X, Y) = 1$ if there exists a subset $\mathcal{X}_r \subseteq \mathcal{X}$ such that for a pair of continuous random variables $(X, Y) \in \mathcal{X}_r \times \mathcal{Y}$, $Y = f(X)$ for a Borel-measurable and non-constant continuous function $f$.

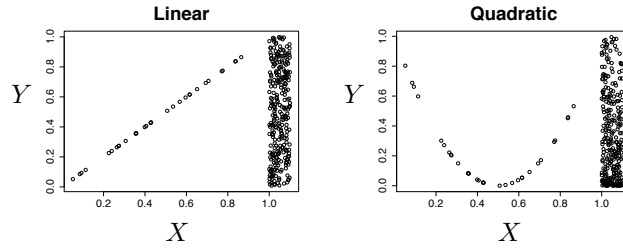

Figure 1: A measure of potential correlation should capture the rare correlation in $X \in [0, 1]$ in these examples which satisfy Axiom 6 for a linear and a quadratic function, respectively.

Axioms 1-5 are identical to a subset of the celebrated axioms of Rényi in [4], which ensure that the measure is properly normalized and invariant under bijective transformations, and recovers the Pearson correlation for jointly Gaussian random variables. Rényi's original axioms for a *measure of correlation* in [4] included Axioms 1-5 and also that the measure $\rho^*$ of correlation should satisfy

6'. $\rho^*(X, Y) = 1$ if for Borel-measurable functions $f$ or $g$, $Y = f(X)$ or $X = g(Y)$.

7'. $\rho^*(X; Y) = \rho^*(Y; X)$.

The Pearson correlation violates a subset (3, 4, and 6') of Rényi's axioms. Together with recent empirical successes in multimodal deep learning (e.g. [26–28]), Rényi's axiomatic approach has been a major justification of Hirschfeld-Gebelein-Rényi (HGR) maximum correlation coefficient defined as $\mathrm{mCor}(X, Y) := \sup_{f,g} \mathsf{E}[f(X)g(Y)]$, which satisfies all Rényi's axioms [2]. Here, the supremum is over all measurable functions with $\mathsf{E}[f(X)] = \mathsf{E}[g(Y)] = 0$ and $\mathsf{E}[f^2(X)] = \mathsf{E}[g^2(Y)] = 1$. However, maximum correlation is not the only measure satisfying all of Rényi's axioms, as we show in the following.

**Proposition 1.** For any function $F : [0, 1] \times [0, 1] \to [0, 1]$ satisfying $F(x, y) = F(y, x)$, $F(x, x) = x$, and $F(x, y) = 0$ only if $xy = 0$, the symmetrized $F(\sqrt{s(X; Y)}, \sqrt{s(Y; X)})$ satisfies all Rényi's axioms.

This follows from the fact that the hypercontractivity coefficient $\sqrt{s(X; Y)}$ satisfies all but the symmetry in Axiom 7 (Theorem 1), and it follows that a symmetrized version satisfies all axioms,

e.g. $(1/2)(\sqrt{s(X;Y)} + \sqrt{s(Y;X)})$ and $(s(X;Y)s(Y;X))^{1/4}$. A formal proof is provided in Appendix A.1.

From the original Rényi's axioms, for potential correlation measure, we remove Axiom 7' that ensures symmetry, as directionality is fundamental in measuring the potential correlation from $X$ to $Y$. We further replace Axiom 6' by Axiom 6, as a variable $X$ has a full potential to be correlated with $Y$ if there exists a domain $\mathcal{X}_r$ such that $X$ and $Y$ are deterministically dependent and non-degenerate (i.e. not a constant function), as illustrated in Figure 1 for a linear function and a quadratic function.

**The hypercontractivity coefficient satisfies all axioms.** We propose the hypercontractivity coefficient $s(X;Y)$, first introduced in [19], as the measure of potential correlation satisfying all Axioms 1-6. Intuitively, $s(X;Y)$ measures how much potential correlation $X$ has with $Y$. For example, if $X$ and $Y$ are independent, then $s(X;Y) = 0$ as $X$ has no correlation with $Y$ (Axiom 3). By data processing inequality, it follows that it is a measure between zero and one (Axiom 2) and also invariant under bijective transformations (Axiom 4). For jointly Gaussian variables $X$ and $Y$ with the Pearson correlation $\rho$, we can show that $s(X;Y) = s(Y;X) = \rho^2$. Hence, the squared-root of $s(X;Y)$ satisfies Axiom 5. In fact, $\sqrt{s(X;Y)}$ satisfies all desired axioms for potential correlation, and we make this precise in the following theorem whose proof is provided in Appendix A.2.

**Theorem 1.** Hypercontractivity coefficient $\sqrt{s(X;Y)}$ satisfies Axioms 1-6.

In particular, the hypercontractivity coefficient satisfies Axiom 6 for potential correlation, unlike other measures of correlation (see Theorem 2 for examples). If there is a potential for $X$ in a possibly rare regime in $\mathcal{X}$ to be fully correlated with $Y$ such that $Y = f(X)$, then the hypercontractivity coefficient is maximum: $s(X;Y) = 1$.

However, just as HGR correlation is not the only one satisfying Rényi's original axioms, the hypercontractivity coefficient is not the only one satisfying our axioms. There is a family of measures known as *hypercontractivity ribbon* that includes the hypercontractivity coefficient as a special case, all of which satisfy the axioms. However, a few properties of the hypercontractivity coefficient makes it more attractive for practical use; it can be efficiently estimated from samples (see Section 3) and is a natural extension of the popular HGR maximal correlation coefficient. Axiom 5 is restricted to univariate $X$ and $Y$, and it can be naturally extended to multivariate variables where $\sqrt{s(X;Y)}$ is a multivariate measure that satisfies all the axioms. For the discussion of hypercontractivity ribbon, connection between hypercontractivity coefficient and HGR maximal correlation, and extension of axioms to multivariate variables, see the journal version [29].

Beside standard correlation measures, another measure widely used to quantify the strength of dependence is mutual information. We can show that mutual information satisfies Axiom 6 if we replace 1 by $\infty$. However there are two key problems: (a) Practically, mutual information is *unnormalized*, i.e., $I(X;Y) \in [0, \infty)$. Hence, it provides no absolute indication of the strength of the dependence. (b) Mathematically, we are looking for a quantity that *tensorizes*, i.e., doesn't change when there are many i.i.d. copies of the same pair of random variables. Hypercontractivity coefficient tensorizes, i.e,

$$s(X_1, ..., X_n; Y_1, .., Y_n) = s(X_1, Y_1), \text{ for i.i.d. } (X_i, Y_i), \ i = 1, \cdots, n.$$

On the other hand, mutual information is additive, i.e.,

$$I(X_1, \cdots, X_n; Y_1, \cdots, Y_n) = nI(X_1; Y_1), \text{ for i.i.d. } (X_i, Y_i), \ i = 1, \cdots, n.$$

Tensorizing quantities capture the strongest relationship among independent copies while additive quantities capture the sum. For instance, mutual information could be large because a small amount of information accumulates over many of the independent components of X and Y (when X and Y are high dimensional) while tensorizing quantities would rule out this scenario, where there is no strong dependence. When the components are not independent, hypercontractivity indeed pools information from different components to find the strongest direction of dependence, which is a desirable property.

One natural way to normalize mutual information is by the log of the cardinality of the input/output alphabets [30]. One can interpret a popular correlation measure MIC as a similar effort for normalizing mutual information and is one of our baselines.

**Standard correlation coefficients violate the Axioms.** We next analyze existing measures of correlations under the scenario with potential correlation (Axiom 6), where we find that none of the

existing correlation measures satisfy Axiom 6. Suppose $X$ and $Y$ are independent (i.e. no correlation) in a subset $\mathcal{X}_d$ of the domain $\mathcal{X}$, and allow $X$ and $Y$ to be arbitrarily correlated in the rest $\mathcal{X}_r$ of the domain, such that $\mathcal{X} = \mathcal{X}_d \cup \mathcal{X}_r$. We further assume that the independent part is dominant and the correlated part is rare; let $\alpha := \mathsf{P}(X \in \mathcal{X}_r)$ and we consider the scenario when $\alpha$ is small. A good measure of potential correlation is expected to capture the correlation in $\mathcal{X}_r$ even if it is rare (i.e., $\alpha$ is small). To make this task more challenging, we assume that the conditional distribution of $Y|\{X \in \mathcal{X}_r\}$ is the same as $Y|\{X \notin \mathcal{X}_r\}$. Figure 1 (of this section) illustrates sampled points for two examples from such a scenario and more examples are in Figure 5 in Appendix B. Our main result is the analysis of HGR maximal correlation (mCor) [2], distance correlation (dCor) [6], maximal information coefficients (MIC) [5], which shows that these measures are vanishing with $\alpha$ even if the dependence in the rare regime is very high. Suppose $Y|(X \in \mathcal{X}_r) = f(X)$, then all three correlation coefficients are vanishing as $\alpha$ gets small. This in particular violates Axiom 6. The reason is that standard correlation coefficients measure the *average correlation* whereas the hypercontractivity coefficient measures the *potential correlation*. The experimental comparisons on the power of these measures confirm our analytical predictions in Figure 2. The formal statement is below and the proof is provided in Appendix A.3.

**Theorem 2.** Consider a pair of continuous random variables $(X, Y) \in \mathcal{X} \times \mathcal{Y}$. Suppose $\mathcal{X}$ is partitioned as $\mathcal{X}_r \cup \mathcal{X}_d = \mathcal{X}$ such that $P_{Y|X}(S|X \in \mathcal{X}_r) = P_{Y|X}(S|X \in \mathcal{X}_d)$ for all $S \subseteq \mathcal{Y}$, and $Y$ is independent of $X$ for $X \in \mathcal{X}_d$. Let $\alpha = \mathsf{P}\{X \in \mathcal{X}_r\}$. The HGR maximal correlation coefficient is

$$\mathrm{mCor}(X, Y) = \sqrt{\alpha}\, \mathrm{mCor}(X_r, Y), \tag{2}$$

the distance correlation coefficient is

$$\mathrm{dCor}(X, Y) = \alpha\, \mathrm{dCor}(X_r, Y), \tag{3}$$

the maximal information coefficient is upper bounded by

$$\mathrm{MIC}(X, Y) \leq \alpha\, \mathrm{MIC}(X_r, Y), \tag{4}$$

where $X_r$ is the random variable $X$ conditioned on the rare domain $X \in \mathcal{X}_r$.

# 3 Estimator of the hypercontractivity coefficient from samples

In this section, we present an algorithm[1] to compute the hypercontractivity coefficient $s(X; Y)$ from i.i.d. samples $\{X_i, Y_i\}_{i=1}^n$. The computation of the hypercontractivity coefficient from samples is known to be challenging for continuous random variables [22, 23], and to the best of our knowledge, there is no known efficient algorithm to compute the hypercontractivity coefficient from samples. Our estimator is the first efficient algorithm to compute the hypercontractivity coefficient, based on the following equivalent definition of the hypercontractivity coefficient, shown recently in [11]:

$$s(X; Y) \equiv \sup_{r_x \neq p_x} \frac{D(r_y \| p_y)}{D(r_x \| p_x)}. \tag{5}$$

There are two main challenges for computing $s(X; Y)$. The first challenge is – given a marginal distribution $r_x$ and samples from $p_{xy}$, how do we estimate the KL divergences $D(r_y \| p_y)$ and $D(r_x \| p_x)$. The second challenge is the optimization over the infinite dimensional simplex. We need to combine estimation and optimization together in order to compute $s(X; Y)$. Our approach is to combine ideas from traditional kernel density estimates and from importance sampling. Let $w_i = r_x(X_i)/p_x(X_i)$ be the *likelihood ratio* evaluated at sample $i$. We propose the estimation and optimization be solved jointly as follows:

**Estimation:** To estimate KL divergence $D(r_x \| p_x)$, notice that

$$D(r_x \| p_x) = \mathbb{E}_{X \sim p_x}\left[\frac{r_x(X)}{p_x(X)} \log \frac{r_x(X)}{p_x(X)}\right].$$

Using empirical average to replace the expectation over $p_x$, we propose

$$\widehat{D}(r_x \| p_x) = \frac{1}{n} \sum_{i=1}^n \frac{r_x(X_i)}{p_x(X_i)} \log \frac{r_x(X_i)}{p_x(X_i)} = \frac{1}{n} \sum_{i=1}^n w_i \log w_i.$$

For $D(r_y || p_y)$, we follow the similar idea, but the challenge is in computing $v_j = r_y(Y_j)/p_y(Y_j)$. To do this, notice that $r_{xy} = r_x p_{y|x}$, so

$$r_y(Y_j) \;=\; \mathbb{E}_{X \sim r_x} \left[ p_{y|x}(Y_j|X) \right] = \mathbb{E}_{X \sim p_x} \left[ p_{y|x}(Y_j|X) \frac{r_x(X)}{p_x(X)} \right] \;.$$

Replacing the expectation by empirical average again, we get the following estimator of $v_j$:

$$\widehat{v}_j \;=\; \frac{1}{n} \sum_{i=1}^{n} \frac{p_{y|x}(Y_j|X_i)}{p_y(Y_j)} \frac{r_x(X_i)}{p_x(X_i)} = \frac{1}{n} \sum_{i=1}^{n} \underbrace{\frac{p_{xy}(X_i, Y_j)}{p_x(X_i)p_y(Y_j)}}_{A_{ji}} w_i \;.$$

We can write this expression in matrix form as $\widehat{\mathbf{v}} = \mathbf{A}^T \mathbf{w}$. We use a kernel density estimator from [31] to estimate the matrix $\mathbf{A}$, but our approach is compatible with any density estimator of choice.

**Optimization:** Given the estimators of the KL divergences, we are able to convert the problem of computing $s(X;Y)$ into an optimization problem over the vector $\mathbf{w}$. Here a constraint of $(1/n)\sum_{i=1}^{n} w_i = 1$ is needed to satisfy $\mathbb{E}_{p_x}[r_x/p_x] = 1$. To improve numerical stability, we use $\log s(X;Y)$ as the objective function.

Then the optimization problem has the following form:

$$\max_{\mathbf{w}} \quad \log \left( (\mathbf{w}^T \mathbf{A} \log(\mathbf{A}^T \mathbf{w})) \right) - \log \left( \mathbf{w}^T \log \mathbf{w} \right)$$
$$\text{subject to} \quad \frac{1}{n} \sum_{i=1}^{n} w_i = 1$$
$$w_i \geq 0, \forall i$$

where $\mathbf{w}^T \log \mathbf{w} = \sum_{i=1}^{n} w_i \log w_i$ for short. Although this problem is not convex, we apply gradient descent to maximize the objective. In practice, we initialize $w_i = 1 + \mathcal{N}(0, \sigma^2)$ for $\sigma^2 = 0.01$. Hence, the initial $r_x$ is perturbed mildly from $p_x$. Although we are not guaranteed to achieve the global maximum, we consistently observe in extensive numerical experiments that we have 50%-60% probability of achieving the same maximum value, which we believed to be the global maximum. A theoretical analysis of the landscape of local and global optima and their regions of attraction with respect to gradient descent is an interesting and challenging open question, outside the scope of this paper. A theoretical understanding of the performance of gradient descent on the optimization step (where the number of samples is fixed) above is technically very challenging and is left to future work.

## 4    Experimental results

We present experimental results on synthetic and real datasets showing that the hypercontractivity coefficient $(a)$ is more powerful in detecting potential correlation compared to existing measures; $(b)$ discovers hidden potential correlations among various national indicators in WHO datasets; and $(c)$ is more robust in discovering pathways of gene interactions from gene expression time series data.

### 4.1    Synthetic data: power test on potential correlation

As our estimator (and the measure itself) involves a maximization, it is possible that we are sensitive to outliers and may capture spurious noise. A formal statistical approach to test the robustness as well as accuracy is to run *power tests*: testing for the power of the estimator in binary hypothesis tests. Via a series of experiments we show that the hypercontractivity coefficient and our estimator are capturing the true potential correlation.

We compare the power of the hypercontractivity coefficient and other correlation coefficients in the binary hypothesis testing scenario of Theorem 2. As shown in Figure 5 in Appendix B, we generate pairs of datasets – one where $X$ and $Y$ are independent and one where there is a potential correlation as per our scenario. We experiment with eight types of functional associations, following the examples from [5, 32, 33]. For the correlated datasets, out of $n$ samples $\{(x_i, y_i)\}_{i=1}^{n}$, $\alpha n$ rare but correlated samples are in $\mathcal{X} = [0, 1]$ and $(1 - \alpha)n$ dominant but independent samples are in $\mathcal{X} \in [1, 1.1]$.

The rare but correlated samples are generated as $x_i \sim \text{Unif}[0, 1], y_i \sim f(x_i) + \mathcal{N}(0, \sigma^2)$ for $i \in [1 : \alpha n]$. The dominant samples are generated as $x_i \sim \text{Unif}[1, 1.1], y_i \sim f(\text{Unif}[0, 1]) + \mathcal{N}(0, \sigma^2)$ for $i \in [\alpha n + 1, n]$. A formal comparison is done via testing their powers: comparing the false negative rate at a fixed false positive rate of, say, 5%. We show empirically that for linear, quadratic, sine with period 1/2, and the step function, the hypercontractivity coefficient is more powerful as compared to other measures. For a given setting, a larger power means a larger successful detection rate for a fixed false alarm rate. Figure 2 shows the power of correlation estimators as a function of the additive noise level, $\sigma^2$, for $\alpha = 0.05$ and $n = 320$. The hypercontractivity coefficient is more powerful than other correlation estimators for most functions. The power of all the estimators are very small for sine (period 1/8) and circle functions. This is not surprising given that it is very hard to discern the correlated and independent cases even visually, as shown in Figure 5. We give extensive experimental results in the journal version [29].

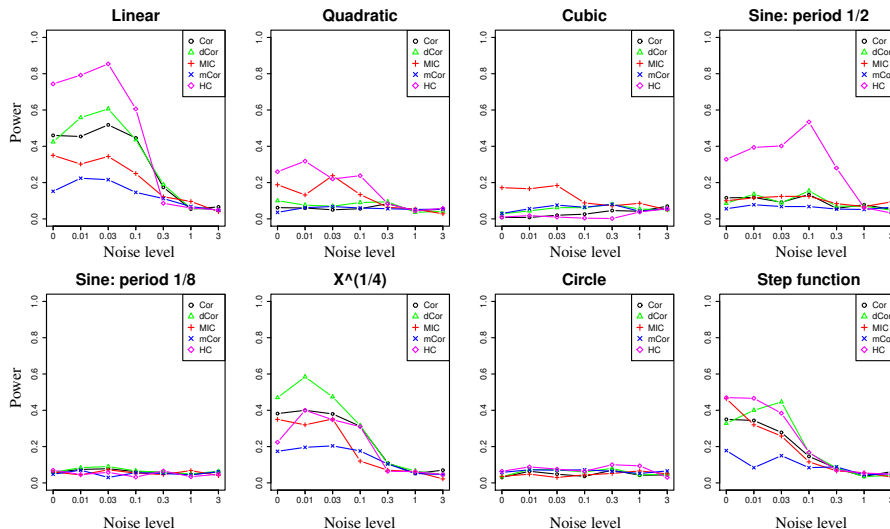

Figure 2: Power vs. noise level for $\alpha = 0.05$, $n = 320$

## 4.2 Real data: correlation between indicators of WHO datasets

We compute the hypercontractivity coefficient, MIC, and Pearson correlation of 1600 pairs of indicators for 202 countries in the World Health Organization (WHO) dataset [5]. Figure 3 illustrates that the hypercontractivity coefficient discovers hidden potential correlation (e.g. in (E) and (F)), whereas other measures fail. Scatter plots of Pearson correlation vs. the hypercontractivity coefficient and MIC vs. the hypercontractivity coefficient for all pairs are presented in Figure 3 (A) and (D). The samples for pairs of indicators corresponding to B,C,E,F in Figure 3 (A) and (D) are shown in Figure 3 (B),(C),(E),(F), respectively. In (B), it is reasonable to assume that the number of bad teeth per child is uncorrelated with the democracy score. The hypercontractivity coefficient, MIC, and Pearson correlation are all small, as expected. In (C), the correlation between $CO_2$ emissions and energy use is clearly visible, and all three correlation estimates are close to one.

However, only the hypercontractivity coefficient discovers the hidden potential correlation in (E) and (F). In (E), the data is a mixture of two types of countries – one with small amount of aid received (less than $\$5 \times 10^8$), and the other with large amount of aid received (larger than $\$5 \times 10^8$). Dominantly many countries (104 out of 146) belong to the first type (small aid), and for those countries, the amount of aid received and the income growth are independent. For the remaining countries with larger aid received, although those are rare, there is a clear correlation between the amount of aid received and the income growth. Similarly in (F), there are two types of countries – one with small arms exports (less than $\$2 \times 10^8$) and the other with large arms exports (larger than $\$2 \times 10^8$). Dominantly many countries (71 out of 82) belong to the first type, for which the amount of arms exports and the health expenditure are independent. For the remaining countries that belong to the second type, on the other hand, there is a visible correlation between the arms exports and the health expenditure. This is expected as for those countries that export arms the GDP is positively correlated

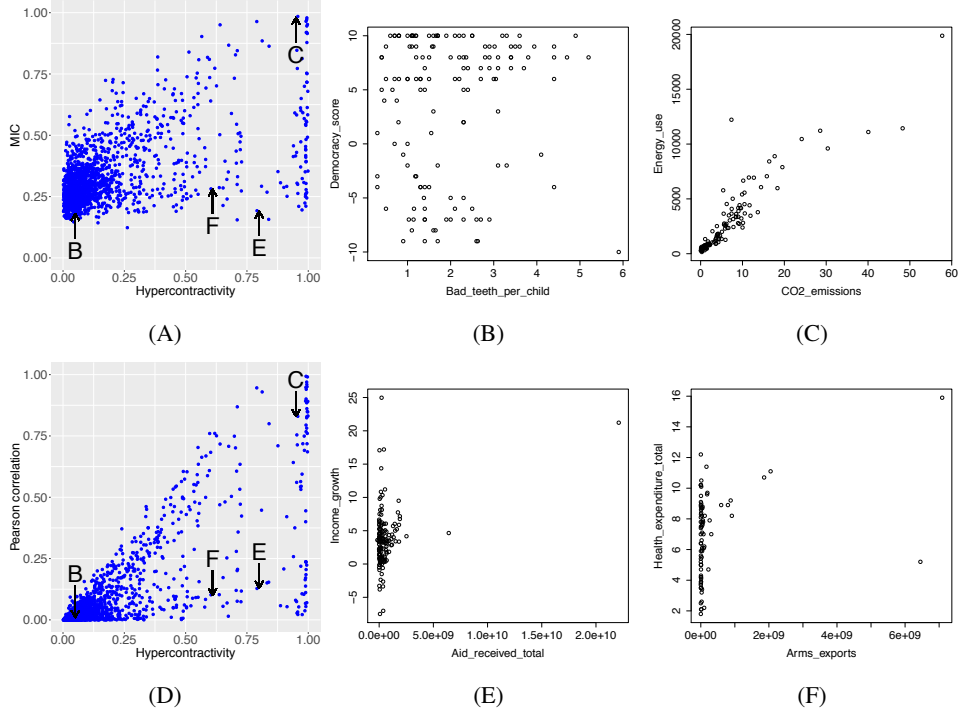

Figure 3: (A) and (D): Scatter plot of correlation measures. (B): Correlations are small. (C): Correlations are large. (E) and (F): Only the hypercontractivity coefficient discovers potential correlation.

with both arms exports and health expenditure, whereas for those do not have arms industry, these two will be independent. We give extensive numerical analyses of the WHO dataset in the journal version [29].

### 4.3 Gene pathway recovery from single cell data

We replicate the genetic pathway detection experiment from [7], and show that hypercontractivity correctly discovers the genetic pathways from smaller number of samples. A genetic pathway is a series of genes interacting with each other as a chain. Consider the following setup where four genes whose expression values in a single cell are modeled by random processes $X_t$, $Y_t$, $Z_t$ and $W_t$ respectively. These 4 genes interact with each other following a pathway $X_t \rightarrow Y_t \rightarrow Z_t \rightarrow W_t$; it is biologically known that $X_t$ causes $Y_t$ with a negligible delay, and later at time $t'$, $Y_{t'}$ causes $Z_{t'}$, and so on. Our goal is to recover this known gene pathway from sampled data points. For a sequence of time points $\{t_i\}_{i=0}^m$, we observe $n_i$ i.i.d. samples $\{X_{t_i}^{(j)}, Y_{t_i}^{(j)}, Z_{t_i}^{(j)}, W_{t_i}^{(j)}\}_{j=1}^{n_i}$ generated from the random process $P(X_{t_i}, Y_{t_i}, Z_{t_i}, W_{t_i})$. We use the real data obtained by the single-cell mass flow cytometry technique [7].

Given these samples from time series, the goal of [7] is to recover the direction of the interaction along the known pathway using correlation measures as follows, where they proposed a new measure called DREMI. The DREMI correlation measure is evaluated on each pairs on the pathway, $\tau(X_{t_i}, Y_{t_i})$, $\tau(Y_{t_i}, Z_{t_i})$ and $\tau(Z_{t_i}, W_{t_i})$, at each time points $t_i$. It is declared that a genetic pathway is correctly recovered if the peak of correlation follows the expected trend: $\arg\max_{t_i} \tau(X_{t_i}, Y_{t_i}) \leq \arg\max_{t_i} \tau(Y_{t_i}, Z_{t_i}) \leq \arg\max_{t_i} \tau(Z_{t_i}, W_{t_i})$. In [25], the same experiment has been done with $\tau$ evaluated by UMI and CMI estimators. In this paper, we evaluate $\tau$ using our proposed estimator of hypercontractivity.

We subsample the raw data from [7] to evaluate the ability to find the trend from smaller samples. Precisely, given a resampling rate $\gamma \in (0, 1]$, we randomly select a subset of indices $S_i \subseteq [n_i]$ with $\text{card}(S_i) = \lceil \gamma n_i \rceil$, compute $\tau(X_{t_i}, Y_{t_i})$, $\tau(Y_{t_i}, Z_{t_i})$ and $\tau(Z_{t_i}, W_{t_i})$ from sub-

samples $\{X_{t_i}^{(j)}, Y_{t_i}^{(j)}, Z_{t_i}^{(j)}, W_{t_i}^{(j)}\}_{j \in S_i}$, and determine whether we can recover the trend successfully, i.e., whether $\arg\max_{t_i} \tau(X_{t_i}, Y_{t_i}) \leq \arg\max_{t_i} \tau(Y_{t_i}, Z_{t_i}) \leq \arg\max_{t_i} \tau(Z_{t_i}, W_{t_i})$. We repeat the experiment several times with independent subsamples and compute the probability of successfully recovering the trend. Figure 4 illustrates that when the entire dataset is available, all methods are able to recover the trend correctly. When only fewer samples are available, hypercontractivity improves upon other competing measures in recovering the hidden chronological order of interactions of the pathway. For completeness, we run datasets for both regular T-cells (shown in left figure) and T-cells exposed with an antigen (shown right figure), for which we expect distinct biological trends. Hypercontractivity method can capture the trend for both datasets correctly and sample-efficiently.

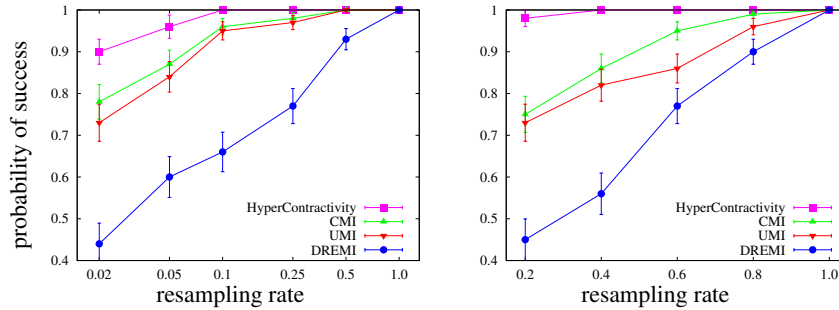

Figure 4: Accuracy vs. subsampling rate. Hypercontractivity method has higher probability to recover the trend when data size is smaller compared to other methods. Left: regular T-cells. Right: T-cells exposed with an antigen [7].

## Acknowledgments

This work was partially supported by NSF grants CNS-1527754, CNS-1718270, CCF-1553452, CCF-1617745, CCF-1651236, CCF-1705007, and GOOGLE Faculty Research Award.

## Footnotes

[1]Code is available at https://github.com/wgao9/hypercontractivity

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
