[Supplementary Material]

# Supplementary Material

## Appendix

## A   Proofs

In this section, we provide proofs for our main results and technical lemmas.

### A.1   Proof of Proposition 1

Let $S_F(X,Y) = F(\sqrt{s(X;Y)}, \sqrt{s(Y;X)})$ for $F$ satisfying conditions in Proposition 1. We show that $S_F(X,Y)$ satisfies all Rényi's axioms, i.e., Axioms 1-5 and 6' and 7'.

1. $S_F(X,Y)$ is defined for any pair of non-constant random variables $X,Y$ because $s(X;Y) \in [0,1]$ and $s(Y;X) \in [0,1]$ are defined for any random variables $X,Y$ by Theorem 1.

2. $S_F(X,Y) \in [0,1]$ because the output of a function $F$ is in $[0,1]$ by the condition on $F$.

3. If $X$ and $Y$ are statistically independent, $s(X;Y) = s(Y;X) = 0$. By the condition on $F$, it follows that $S_F(X,Y) = 0$. If $S_F(X,Y) = 0$, by the condition on $F$, $s(X;Y)s(Y;X) = 0$, which implies that $X$ and $Y$ are statistically independent.

4. $S_F(f(X), g(Y)) = S_F(X,Y)$ for any bijective Borel-measurable functions $f, g$ because $\sqrt{s(f(X);g(Y))} = \sqrt{s(X;Y)}$ and $\sqrt{s(g(Y);f(X))} = \sqrt{s(Y;X)}$ by Theorem 1.

5. For $(X,Y) \sim \mathcal{N}(\mu, \Sigma)$ with Pearson correlation $\rho$, $s(X;Y) = s(Y;X) = \rho^2$. Hence, $S_F(X,Y) = F(|\rho|, |\rho|) = |\rho|$.

6' If $Y = f(X)$ for a non-constant function $f$, it follows that $I(f(X); f(X)) = I(f(X); X)$ because if $f(X)$ is discrete, $I(f(X); f(X)) = I(f(X); X) = H(f(X))$ and otherwise, $I(f(X); f(X)) = I(f(X); X) = \infty$. Hence

$$s(X; f(X)) = \sup_{U - X - f(X)} I(U; f(X))/I(U;X) = I(f(X); f(X))/I(f(X); X) = 1.$$

   Similarly, $s(f(X); X) = \sup_{U - f(X) - X} I(U;X)/I(U;f(X)) = 1$. Hence, $S_F(X; f(X)) = F(1,1) = 1$. Likewise, we can show that $S_F(X;Y) = 1$ if $X = g(Y)$.

7' $S_F(X,Y) = S_F(Y,X)$ because $F(x,y) = F(y,x)$.

### A.2   Proof of Theorem 1

We show that $s(X;Y)$ satisfies Axioms 1-6 in Section 2.

1. For any non-constant random variable $X$, $\exists U$ s.t. $I(U;X) > 0$. Hence, $s(X;Y)$ is defined for any pair of non-constant random variables $X$ and $Y$.

2. Since mutual information is non-negative, $s(X;Y) \geq 0$. By data processing inequality, for any $U - X - Y$, $I(U;X) \leq I(U;Y)$. Hence, $s(X;Y) \leq 1$.

3. If $X$ and $Y$ are independent, for any $U$, $I(U;Y) \leq I(X;Y) = 0$. Hence, $s(X;Y) = 0$. If $X$ and $Y$ are dependent, $I(X;Y) > 0$, which implies that $s(X;Y) \geq I(X;Y)/H(X) > 0$.

4. For any bijective functions $f, g$,

$$I(U; g(Y)) = I(U; g(Y), Y) = I(U;Y) + I(U; g(Y)|Y) = I(U;Y).$$

   Similarly, $I(U; f(X)) = I(U;X)$. Hence,

$$
\begin{aligned}
s(f(X); g(Y)) &= \sup_{U: U - f(X) - g(Y), I(U;f(X)) > 0} \frac{I(U; g(Y))}{I(U; f(X))} \\
&= \sup_{U: U - X - f(X) - g(Y) - Y, I(U;X) > 0} \frac{I(U;Y)}{I(U;X)} \\
&= s(X;Y).
\end{aligned}
$$

5. By Theorem 3.1 in [34], for $(X, Y)$ jointly Gaussian with correlation coefficient $\rho$,

$$\min_{U:\, U-X-Y} (I(U; X) - \beta I(U; Y)) = 0$$

for $\beta \le 1/\rho^2$. Equivalently,

$$\max_{U:\, U-X-Y} \left(I(U; Y) - \rho^2 I(U; X)\right) = 0,$$

which implies that $s(X; Y) \le \rho^2$. To show that $s(X; Y) \ge \rho^2$, let $U_Z = X + Z$ for $Z \sim (0, \sigma_1^2)$. Consider

$$
\begin{aligned}
s(X; Y) &\ge \lim_{\sigma_1^2 \to \infty} \frac{I(U_Z; Y)}{I(U_Z; X)} \\
&= \lim_{\sigma_1^2 \to \infty} \frac{\log\left(\frac{(\sigma_X^2 + \sigma_1^2)\sigma_Y^2}{(\sigma_X^2 + \sigma_1^2)\sigma_Y^2 - \rho^2 \sigma_X^2 \sigma_Y^2}\right)}{\log\left(1 + \frac{\sigma_X^2}{\sigma_1^2}\right)} \\
&= \lim_{\sigma_1^2 \to \infty} \frac{\rho^2 \sigma_X^2 \sigma_Y^2 / \left((\sigma_X^2 + \sigma_1^2)\sigma_Y^2 - \rho^2 \sigma_X^2 \sigma_Y^2\right)}{\sigma_X^2 / \sigma_1^2} \\
&= \rho^2.
\end{aligned}
$$

Hence, $s(X; Y) = \rho^2$. An alternative proof is provided in [24].

6. To prove that $s(X; Y)$ satisfies Axiom 6, we first show the following lemma.

**Lemma 1.** Consider a pair of random variables $(X, Y) \in \mathcal{X} \times \mathcal{Y}$. The hypercontractivity $s(X; Y)$ is lower bounded by

$$s(X; Y) \ge \frac{I(U; Y | X \in \mathcal{X}_r)}{H(\alpha)/\alpha + I(U; X | X \in \mathcal{X}_r)} \tag{6}$$

for any $\mathcal{X}_r$ such that $\mathcal{X}_r \subseteq \mathcal{X}$ for $\mathsf{P}\{X \in \mathcal{X}_r\} =: \alpha > 0$.

*Proof.* Let

$$U_s = \begin{cases} U \sim p(u|x) & \text{if } X \in \mathcal{X}_r, \\ \emptyset & \text{otherwise.} \end{cases} \tag{7}$$

Let $S = \mathbb{I}_{\{U_s = \emptyset\}} = \mathbb{I}_{\{X \in \mathcal{X}_r\}}$. Note that $S - U_s - X - Y$ holds, and that $S$ is a deterministic function of $X$. Hence,

$$
\begin{aligned}
I(U_s; X) &= I(U_s, S; X) \\
&= I(S; X) + I(U_s; X | S) \\
&= H(\alpha) + \alpha I(U; X | X \in \mathcal{X}_r). \tag{8}
\end{aligned}
$$

Consider

$$
\begin{aligned}
I(U_s; Y) &= I(U_s, S; Y) \\
&= I(S; Y) + I(U_s; Y | S) \\
&\ge \alpha I(U; Y | X \in \mathcal{X}_r). \tag{9}
\end{aligned}
$$

The proof is completed by combining (8) and (9). $\qquad \square$

Assume that $Y = f(X)$ for $X \in \mathcal{X}_r$. Considering $U = f(X)$ in (7) in Lemma 1, we obtain the following lower bound:

$$s(X; Y) \ge \frac{I(f(X); f(X) | X \in \mathcal{X}_r)}{H(\alpha)/\alpha + I(f(X); X | X \in \mathcal{X}_r)}.$$

For any continuous random variable $X$ and a non-constant continuous function $f$, $I(f(X); f(X) | X \in \mathcal{X}_r) = I(f(X); X | X \in \mathcal{X}_r) = \infty$, which implies that $s(X; Y) = 1$.

## A.3 Proof of Theorem 2

We first prove that $\mathrm{mCor}(X,Y) = \sqrt{\alpha}\,\mathrm{mCor}(X_r,Y)$ in (2). Let $S = \mathbb{I}_{\{X \in \mathcal{X}_r\}}$ be the indicator for whether $X \in \mathcal{X}_r$ or not. Consider

$$
\begin{aligned}
\mathrm{mCor}(X;Y) &= \max_{\substack{f,g \\ :\mathsf{E}[f(X)]=\mathsf{E}[g(Y)]=0, \\ \mathsf{E}[f^2(X)]\le 1, \mathsf{E}[g^2(Y)]\le 1}} \mathsf{E}[f(X)g(Y)] \\[4pt]
&= \max_{\substack{f,g \\ :\mathsf{E}[f(X)]=\mathsf{E}[g(Y)]=0, \\ \mathsf{E}[f^2(X)]\le 1, \mathsf{E}[g^2(Y)]\le 1}} \mathsf{E}_S[\mathsf{E}[f(X)g(Y)|S]] \\[4pt]
&= \max_{\substack{f,g \\ :\mathsf{E}[f(X)]=\mathsf{E}[g(Y)]=0, \\ \mathsf{E}[f^2(X)]\le 1, \mathsf{E}[g^2(Y)]\le 1}} (\alpha\,\mathsf{E}[f(X)g(Y)|X \in \mathcal{X}_r] + \bar{\alpha}\,\mathsf{E}[f(X)g(Y)|X \in \mathcal{X}_d]) \\[4pt]
&= \max_{\substack{f,g \\ :\mathsf{E}[f(X)]=\mathsf{E}[g(Y)]=0, \\ \mathsf{E}[f^2(X)]\le 1, \mathsf{E}[g^2(Y)]\le 1}} (\alpha\,\mathsf{E}[f(X)g(Y)|X \in \mathcal{X}_r] + \bar{\alpha}\,\mathsf{E}[f(X)|X \in \mathcal{X}_d]\,\mathsf{E}[g(Y)|X \in \mathcal{X}_d]) \\[4pt]
&\overset{(a)}{=} \alpha \max_{\substack{f,g \\ :\mathsf{E}[f(X)]=\mathsf{E}[g(Y)]=0, \\ \mathsf{E}[f^2(X)]\le 1, \mathsf{E}[g^2(Y)]\le 1}} \mathsf{E}[f(X)g(Y)|X \in \mathcal{X}_r] \\[4pt]
&\overset{(b)}{=} \sqrt{\alpha}\,\mathrm{mCor}(X_r,Y).
\end{aligned}
$$

Step $(a)$ holds since $\mathsf{E}[g(Y)|X \in \mathcal{X}_r] = \mathsf{E}[g(Y)|X \in \mathcal{X}_d]$ from the assumption that marginal distributions are equal, and that $\mathsf{E}[g(Y)] = \alpha\,\mathsf{E}[g(Y)|X \in \mathcal{X}_r] + \bar{\alpha}\,\mathsf{E}[g(Y)|X \in \mathcal{X}_d]$. To show step $(b)$, let $c = \mathsf{E}[f(X)|X \in \mathcal{X}_d]$ and note that

$$
\begin{aligned}
\alpha\,\mathsf{E}[f(X)|X \in \mathcal{X}_r] &= -\bar{\alpha}c, \\
\alpha\,\mathsf{E}[f^2(X)|X \in \mathcal{X}_r] &= \mathsf{E}[f^2(X)] - \bar{\alpha}\,\mathsf{E}[f^2(X)|X \in \mathcal{X}_d] \\
&\le 1 - \bar{\alpha}c^2, \\
\mathsf{E}[g(Y)|X \in \mathcal{X}_r] &= 0.
\end{aligned}
$$

Hence,

$$
\begin{aligned}
\max_{\substack{f,g \\ :\mathsf{E}[f(X)]=\mathsf{E}[g(Y)]=0, \\ \mathsf{E}[f^2(X)]\le 1, \mathsf{E}[g^2(Y)]\le 1}} \mathsf{E}[f(X)g(Y)|X \in \mathcal{X}_r] &= \max_{\substack{f_r,g \\ :\mathsf{E}[f_r(X)]=-\bar{\alpha}c/\alpha, \mathsf{E}[g(Y)]=0, \\ \mathsf{E}[f_r^2(X)]\le(1-\bar{\alpha}c^2)/\alpha, \mathsf{E}[g^2(Y)]\le 1}} \mathsf{E}[f_r(X)g(Y)] \\[4pt]
&= \max_{\substack{f_{rc},g \\ :\mathsf{E}[f_{rc}(X)]=0, \mathsf{E}[g(Y)]=0, \\ \mathsf{E}[f_{rc}^2(X)]\le(\alpha-\bar{\alpha}c^2)/\alpha^2, \mathsf{E}[g^2(Y)]\le 1}} \mathsf{E}[(f_{rc}(X)g(Y)] \\[4pt]
&= \max_{\substack{f_{rc},g \\ :\mathsf{E}[f_{rc}(X)]=0, \mathsf{E}[g(Y)]=0, \\ \mathsf{E}[f_{rc}^2(X)]\le 1/\alpha, \mathsf{E}[g^2(Y)]\le 1}} \mathsf{E}[f_{rc}(X)g(Y)] \\[4pt]
&= \max_{\substack{f_{rca},g \\ :\mathsf{E}[f_{rca}(X)]=0, \mathsf{E}[g(Y)]=0, \\ \mathsf{E}[f_{rca}^2(X)]\le 1, \mathsf{E}[g^2(Y)]\le 1}} \frac{1}{\sqrt{\alpha}}\,\mathsf{E}[f_{rca}(X)g(Y)] \\[4pt]
&= \frac{\mathrm{mCor}(X_r,Y)}{\sqrt{\alpha}},
\end{aligned}
$$

where $f_r(X)$, $f_{rc}(X) = f_r(X) + \bar{\alpha}c/\alpha$, and $f_{rca}(X) = \sqrt{\alpha}f_{rc}(X)$ are functions defined only for $X \in \mathcal{X}_r$.

We next show $\mathrm{dCor}(X,Y) = \alpha\,\mathrm{dCor}(X_r,Y)$ in (3). Let

$$
h_X(s) = \mathsf{E}[e^{isX}],\ h_Y(t) = \mathsf{E}[e^{itY}],\ h_{XY}(s,t) = \mathsf{E}[e^{i(sX+tY)}].
$$

Note that

$$h_{XY}(s,t) = \mathsf{E}[e^{i(sX+tY)}]$$
$$= \alpha\,\mathsf{E}[e^{i(sX+tY)}|X\in\mathcal{X}_r] + \bar{\alpha}\,\mathsf{E}[e^{isX}|X\in\mathcal{X}_d]\,\mathsf{E}[e^{itY}|X\in\mathcal{X}_d]$$
$$= \alpha\,\mathsf{E}[e^{i(sX+tY)}|X\in\mathcal{X}_r] + \bar{\alpha}\,\mathsf{E}[e^{isX}|X\in\mathcal{X}_d]\,\mathsf{E}[e^{itY}], \tag{10}$$

and

$$h_X(s) = \mathsf{E}[e^{isX}] = \alpha\,\mathsf{E}[e^{isX}|X\in\mathcal{X}_r] + \bar{\alpha}\,\mathsf{E}[e^{isX}|X\in\mathcal{X}_d]. \tag{11}$$

By combining (10) and (11),

$$h_{XY}(s,t) - h_X(s)h_Y(t) = \alpha\,\mathsf{E}[e^{i(sX+tY)}|X\in\mathcal{X}_r] - \alpha\,\mathsf{E}[e^{isX}|X\in\mathcal{X}_r]\,\mathsf{E}[e^{itY}]$$
$$= \alpha\,\mathsf{E}[e^{i(sX+tY)}|X\in\mathcal{X}_r] - \alpha\,\mathsf{E}[e^{isX}|X\in\mathcal{X}_r]\,\mathsf{E}[e^{itY}|X\in\mathcal{X}_r]$$
$$= \alpha\,\mathrm{dCor}(X_r,Y).$$

Finally, we show that $\mathrm{MIC}(X,Y) \le \alpha\,\mathrm{MIC}(X_r,Y)$ in (4).

Let $X_Q(X) \in \mathcal{X}_Q(X)$ and $Y_Q(Y) \in \mathcal{Y}_Q(Y)$ denote a quantization of $X$ and $Y$, respectively. Consider

$$\mathrm{MIC}(X,Y) = \max_{X_Q(X),Y_Q(Y)} \frac{I(X_Q;Y_Q)}{\log\min\{|\mathcal{X}_Q|,|\mathcal{Y}_Q|\}}$$
$$\le \max_{X_Q(X),Y_Q(Y)} \frac{I(\mathbb{I}_{X\in\mathcal{X}_r},X_Q;Y_Q)}{\log\min\{|\mathcal{X}_Q|,|\mathcal{Y}_Q|\}}$$
$$\overset{(a)}{=} \alpha \max_{X_Q(X),Y_Q(Y)} \frac{I(X_Q;Y_Q|X\in\mathcal{X}_r)}{\log\min\{|\mathcal{X}_Q|,|\mathcal{Y}_Q|\}}$$
$$\le \alpha \max_{X_Q(X_r),Y_Q(Y)} \frac{I(X_Q;Y_Q|X\in\mathcal{X}_r)}{\log\min\{|\mathcal{X}_Q(X_r)|,|\mathcal{Y}_Q|\}}$$
$$= \alpha\,\mathrm{MIC}(X_r,Y),$$

where step $(a)$ holds because $\mathbb{I}_{X\in\mathcal{X}_r}\!\perp\!\!\!\perp Y$ implies $\mathbb{I}_{X\in\mathcal{X}_r}\!\perp\!\!\!\perp Y_Q$ and $X\!\perp\!\!\!\perp Y$ in $X\in\mathcal{X}_d$ implies $X_Q\!\perp\!\!\!\perp Y_Q$ in $X\in\mathcal{X}_d$.

## B  Supplementary to synthetic experiments

We provide additional experimental results for the synthetic data considered in Section 4.1. Figure 5 shows examples of correlated (right) and independent (left) datasets generated as in Section 4.1; for the correlated datasets on the right, out of $n$ samples $\{(x_i,y_i)\}_{i=1}^n$, $\alpha n$ rare but correlated samples are in $\mathcal{X}=[0,1]$ and $(1-\alpha)n$ dominant but independent samples are in $\mathcal{X}\in[1,1.1]$. The rare but correlated samples are generated as $x_i\sim\mathrm{Unif}[0,1]$, $y_i\sim f(x_i)+\mathcal{N}(0,\sigma^2)$ for $i\in[1:\alpha n]$. The dominant samples are generated as $x_i\sim\mathrm{Unif}[1,1.1]$, $y_i\sim f(\mathrm{Unif}[0,1])+\mathcal{N}(0,\sigma^2)$ for $i\in[\alpha n+1,n]$.

Table 1 shows the hypercontractivity coefficient and the other correlation coefficients for correlated and independent datasets shown in Figure 5, along with the chosen value of $\alpha$ and $\sigma^2$. Correlation estimates with the largest separation for each row is shown in bold. The hypercontractivity coefficient gives the largest separation between the correlated and the independent dataset for most functional types.

To compute the power of each estimator, we generate 500 independent datasets and 500 correlated datasets. We compute the correlation estimates on 500 independent samples, and take the top 5% as a threshold. We compute the correlation estimates on 500 correlated samples. Power is defined as the fraction of correlated datasets for which the correlation estimate is larger than the threshold. We compare the power of hypercontractivity and other estimators for various values of $\alpha$, $\sigma^2$ and $n$.

## C  Supplementary to the experiments for gene pathway data

We provide more details about the gene pathway detection experiment here. The Figure 6 shows the scatter plots pCD3$\zeta$-pSLP76-pERK-pS6 chain at different time points after TCR activation. The data

Figure 5: Sample data points for eight functions with/without a potential correlation for $n = 320$.

Table 1: Comparison of correlation coefficients for independent and correlated samples from Figure 5

| # | Function | $\alpha$ | $\sigma^2$ | Cor dep | Cor indep | dCor dep | dCor indep | mCor dep | mCor indep | MIC dep | MIC indep | HC dep | HC indep |
|---|----------|----------|-----------|---------|-----------|----------|------------|----------|------------|---------|-----------|--------|----------|
| 1 | Linear | 0.05 | 0.03 | 0.03 | 0.00 | 0.19 | 0.11 | 0.06 | 0.04 | 0.21 | 0.17 | **0.18** | **0.08** |
| 2 | Quadratic | 0.10 | 0.10 | 0.00 | 0.01 | 0.09 | 0.10 | **0.07** | **0.02** | 0.21 | 0.18 | 0.08 | 0.04 |
| 3 | Cubic | 0.10 | 0.00 | 0.02 | 0.00 | 0.16 | 0.08 | 0.09 | 0.03 | **0.26** | **0.17** | 0.11 | 0.04 |
| 4 | $\sin(4\pi X)$ | 0.05 | 0.03 | 0.00 | 0.00 | 0.10 | 0.06 | 0.03 | 0.01 | 0.20 | 0.18 | **0.10** | **0.04** |
| 5 | $\sin(16\pi X)$ | 0.10 | 0.00 | 0.00 | 0.00 | 0.07 | 0.08 | **0.03** | **0.03** | 0.18 | 0.22 | **0.03** | **0.03** |
| 6 | $X^{1/4}$ | 0.05 | 0.01 | 0.01 | 0.00 | 0.12 | 0.07 | 0.02 | 0.01 | 0.20 | 0.20 | **0.12** | **0.04** |
| 7 | Circle | 0.10 | 0.00 | 0.00 | 0.00 | 0.09 | 0.05 | 0.01 | 0.03 | 0.16 | 0.17 | **0.06** | **0.01** |
| 8 | Step func. | 0.10 | 0.03 | 0.00 | 0.00 | 0.13 | 0.07 | 0.04 | 0.02 | 0.20 | 0.17 | **0.11** | **0.04** |

comes from CD4+ naïve T lymphocytes from B6 mice with CD3, CD28, and CD4 cross-linking. Each row represents a pair of data in the chain, and each column stands for a time point after TCR activation. Estimate of hypercontractivity is shown below the scatter plot for each pair of data and each time point and we highlight the time point where each pair of data is maximally correlated. We can see that the peak of the correlation of pCD3$\zeta$-pSLP76, pSLP76-pERK and pERK-pS6 appears at 0.5 min, 1 min and 2 min respectively, hence the pathway is correctly identified.

In Figure 7, the similar plots was shown for T-cells exposed with an antigen. Similarly, hypercontractivity is able to capture the trend.

Figure 6: Scatter plots of gene pathway data for various pair of data and various time points (regular T-cells).

Figure 7: Scatter plots of gene pathway data for various pair of data and various time points (T-cells exposed with an antigen).