[Reviews · NeurIPS 2017]

Reviewer 1



This paper suggests a measure of “potential influence” between two random variables, and a method for estimating this measure from data samples. The motivation is that existing measures are "average" measures, and cannot capture “rare” but important dependencies. Specifically, the setup considered by the authors is where two random variables X and Y are independent on much of X’s support (or domain), but there is a small subset of X on which Y and X have strong dependency. This is articulated in the Axiom 6 in the paper, which requires that a proper measure of influence should attain a value 1, if there is a sub-domain of the variable over which Y=f(X) is a deterministic (Borel measurable and non-constant) function. This is a well-written paper on an interesting topic. The proposed method for estimating hypercontractivity coefficient looks interesting. The authors conduct experiments with both synthetic and real-world data and show relative advantage of the proposed measure over some alternatives. At the same time, it’s not clear to me what are the advantages of the proposed measure over mutual information (MI). In fact, it seems that MI can capture similar type of dependency, and one only have to modify the 6th axiom, so that instead of 1 the measure would be infinity. One practical issue for using MI as a measure of dependency is that most existing estimators fail to capture strong relationships. This was pointed out in [Gao et. el, AISTATS’15, “Efficient Estimation of Mutual Information for Strongly Dependent Variables”, https://arxiv.org/abs/1411.2003], where the authors showed that, for a wide class of estimators, accurate estimation of mutual information, I(X:Y) between two variables X and Y, requires a sample size that is exponential in I(X:Y), due to strong boundary effects. Thus, when Y is a near-deterministic function of X (so that I(X:Y) diverges), one needs extremely large sample size for accurate estimation. This point was also shown empirically in a recent critique of MIC by Kinney and G. Atwal, Equitability, mutual information, and the maximal information coefficient, PNAS 111(9):3354–3359, 2014. Furthermore, the Gao et. al. paper also proposed a new estimator that corrected for the boundary bias, and which was shown to capture strong relationships with limited samples. Thus, to claim that the hypercontractivity coefficient is a better measure than MI, the authors should compare their results to Gao et. al.’s estimator. Other comments: - The authors have to clarify the setup more explicitly, e.g., that they deal with one dimensional variables. For instance, the main intuition stated in the Introduction is that “if X and Y are highly correlated, it is possible to get a very compact summary of X that is very informative about Y”. It’s obvious that this statement do not hold in the most general case [e.g., if Y is a binary variable, X is a p-dimensional real vector where each component is a noisy copy of Y, then X can be strongly correlated with X, yet any compression of X will lead to information loss] - The connection with IB is not very clear, especially since the authors do not specify the set up. E.g., in typical IB setting, X is a high dimensional variable, Y is a label (from a finite set), and U is a compression of X. Looking at the objective in Eq. (1), wouldn’t it be maximized if U was an exact copy of Y? - It would help if the supplementary material contained mostly novel contributions, and used references to point to existing relevant work.

Reviewer 2



This paper explores the usage of the hypercontractivity (HC) measure to discover the potential influence of one variable on another. One drawback of previous measures, such as the correlation coefficient, is that they measure only the average influence; whereas sometimes one is also interested in discovering whether there exists any dependence in the data, even one that holds only on a subset of values. The paper proposes a set of desirable axioms (inspired by Renyi's axioms for a measure of correlation) for such a measure, and then proceeds to show that HC fulfills all of them. Furthermore, a method to estimate the HC from samples is presented (also shown to be consistent). Finally, the paper applies the estimator to both synthetic and real-world data, showing that it successfully captures dependencies that make intuitively sense and that other measures do not capture. PROS: - This is a clean and well-executed paper that conceptually clarifies the HC coefficient and its usage in estimating dependencies between two random variables. In particular, axiom 6 (apparently novel) is IMO an extremely powerful requirement. CONS: - None really, apart from the very minor detail that at least 1-2 of the synthetic datasets should have been in the main text for illustrative purposes.

Reviewer 3



Summary: Finding measures of correlation among data covariates is an important problem. There have been several measures that have been proposed in the past - maximum correlation, distance correlation and maximum information coefficient. This paper introduced a new correlation measure by connecting dots among some recent work in information theory literature. The candidate in this paper is called hyper-contractivity coefficient or recently also is known as information bottleneck. Information bottleneck principle is this: Whats the best summary of variable X (most compressed) such that it also retains a lot of information about another variable Y. So for all summaries of X, U the maximum ratio of I(U;Y)/I(U;X) is the hyper contractivity coefficient. This quantity has another interpretation due to recent work. Consider the conditional p(y|x) and marginal p(x) associate with joint distribution of (X,Y). Consider a different distribution r(x) and the marginal r(y) due to the joint r(x) p(y|x). The hypercontractivity coefficient is the maximum ratio of Divergence between r(y) (induced distribution on y by the change distribution r(x)) and p(y) to the divergence between r(x) and p(x). In other words "whats the maximum change at output y that can be caused by a new distribution that looks similar to x in KL distance" Renyi has come up with several axioms for characterizing correlation measures. The authors show that if you make one of the axioms strict in a sense, and then drop the need for symmetry, then hypercontractivity satisfies all the axioms. To be less abstract, the goal of authors is that they like to have a measure which has high correlation value even in for most values of X there seems to be no influence on Y while in some rare region of the X space, correlation with Y is almost deterministic. This has practical applications as the authors demonstrate in the experiments. It turns out the hypercontractivity coefficient has this property while all other previous measures do not have this property provably. This is also related to their modification of the axioms of Renyi by a stronger one (axiom 6 instead of 6'). the authors propose an asymptotically consistent estimator using the second definition of hyper contractivity. They define an optimization problem in terms of the ratio of the change distribution r(x) and original distribution p(x) which they solve by gradient descent.To come to this formulation they use existing results in kernel density estimation + clever tricks in importance sampling and the defn of hyper contractivity. The authors justify the use of the measure by a very exhanustive set of experiments. Strengths: a) The paper connects hyper contractivity ( which has recently acquired attention in information theory community) to a correlation measure that can actually pick out "hidden strong correlations under rare events" much better than the existing ones. Theoretically justifying it with appropriate formulation of axioms based on Renyi's axioms and clarifying relationships amongs various other correlation measures is pretty interesting and a strong aspect of this paper. b) Authors apply this to discover existing known gene pathways from data and also discover potential correlation among WHO indicators which are difficult to detect. There is an exhaustive simulations with synthetic functions too. I have the following concerns too: a) Detecting influence from X to Y is sort of well studied in causality literature. In fact counterfactual scores called probability of sufficiency and probability of necessity have been proposed (Please refer "Causality" book by Judea Pearl 2009. Chapter 7 talks about scores for quantifying influence). So authors must refer to and discuss relationships to such counterfactual measures. Since these are counterfactual measures, they also do not measure just factual influence. Further , under some assumptions the above counterfactual measures can be computed from observational data too. Such key concepts must be referred to and discussed if not compared. I however admit that the current works focus is on correlation measures and not on influence in the strictest causal sense. b) This comment is related to my previous comment : The authors seem to suggest that hypercontractivity suggest a direction of influence from X to Y ( this suggestion is sort of vague in the paper). I would like to point out two things : a) In experiments in the WHO dataset, some of the influences are not causal (for example case E) and some of them may be due to confounders (like F between arms exports and health expenditure). So this measure is about finding hidden potential correlations or influences if the causal direction if already known b) For gaussians, S(Y,X) and S(X,Y) are same - therefore directional differences depend on the functional relationships between X and Y. So the title of discovering potential influence may not be a suitable title given prior work on counterfactual influence scores? I agree however that it is discovering rare hidden correlations. c) Does it make sense to talk about a conditional version ? If so where would those be useful ? d) Section 4.3 - Authors say "X_t will cause ... Y_t to change at a later point in time and so on". However, the authors for the same t_i check for the measure s(X_{t_i},Y_{t_i}). So are we not measure instantaneous influence here. I agree that the instantaneous influences can come one after the other in a sequence of pairs of genes. However, this discrepancy needs to be clarified. e) Why did the authors not use UMI or CMI estimators for experiments in Section 4.2 and 4.1 It seems like shannon capacity related estimators proposed in ref 26 does loosely have properties like that of axiom 6 ( roughly speaking). Actually I wonder why the authors did not include shannon capacity in the theoretical treatment with other measures (I understand that certain normalization criterions like being between 0 and 1 and 5 are not satisfied)? f)Slightly serious concern: Does Axiom 6, for a given {\cal X,\cal Y} domain need to hold for "some" non constant function f or "every" non constant function f ?? Axiom 6' is for any function f ?? The reason I am asking is because the last expression in page 15 in the appendix has H(f(X) in the numerator and denominator. The authors say "H(f(X)) =infinity for a continuous random variable X and a nonconstant function f" - So clearly if this is the proof Axiom 6 is not for any f. Because f being a linear function and X being a uniform distribution on bounded support clearly does not satisfy that statement in the proof if it means any f. g) Follow up of concerns in (f) : In page 14, proof of Proposition 1, point 6' - If Y=f(X) s(X;f(X)) =1 from Theorem 1" . That seems to rely on the above concern in (f). It seems like axiom 6 is not valid for any f. In that case, how is proposition 1 point 6' correct? I think that needs a separate proof for alpha=1 right ?? h) Regarding directionality: Can you characterize functions f, noise such that Y= f(X, noise) and S (Y,X) << S(X,Y). If this can be done, this could be used potentially for casual inference between pairs from just observational data on which there have been several works in the past few years. Have the authors considered applying this to that case? i) With respect to the material on page 27 in the appendix: It seems like definition in (5) has an interventional interpretation. You are trying different soft interventions r(x) at x and you are checking for what soft intervention that is not too different from p(x) the channel output changes dramatically. This seems like a measure that sort of simulates an intervention assuming a causal direction to check the strength of causation. In this sense, mildly it is a counterfactual measure. Can this be use as a substitute for interventions in certain structural equation models (under some suitable assumptions on non-linearity noise etc). If authors have answers, I would like to hear it.